METHODS AND RESOURCES

# Improved genetically encoded near-infrared fluorescent calcium ion indicators for *in vivo* imaging

Yong Qian[1,2], Danielle M. Orozco Cosio[2], Kiryl D. Piatkevich[2,3], Sarah Aufmkolk[4,5,6], Wan-Chi Su[7], Orhan T. Celiker[2], Anne Schohl[4], Mitchell H. Murdock[2], Abhi Aggarwal[8], Yu-Fen Chang[7], Paul W. Wiseman[5,9], Edward S. Ruthazer[4], Edward S. Boyden[2], Robert E. Campbell[1,10]*

1 Department of Chemistry, University of Alberta, Edmonton, Alberta, Canada, 2 Departments of Biological Engineering, Media Arts and Sciences, Brain and Cognitive Sciences, McGovern Institute, Koch Institute, Center for Neurobiological Engineering, MIT, and Howard Hughes Medical Institute, Cambridge, Massachusetts, United States of America, 3 School of Life Sciences, Westlake University, Hangzhou, China, 4 Montreal Neurological Institute-Hospital, Department of Neurology and Neurosurgery, McGill University, Montreal, Quebec, Canada, 5 Department of Chemistry, McGill University, Montreal, Quebec, Canada, 6 Department of Genetics, Harvard Medical School, Boston, Massachusetts, United States of America, 7 LumiSTAR Biotechnology, Nangang District, Taipei City, Taiwan, 8 Howard Hughes Medical Institute, Janelia Research Campus, Ashburn, Virginia, United States of America, 9 Department of Physics, McGill University, Montreal, Quebec, Canada, 10 Department of Chemistry, The University of Tokyo, Tokyo, Japan

* robert.e.campbell@ualberta.ca

**Data Availability Statement:** All relevant data are within the paper and its Supporting information. Plasmids pAAV-CAG-NIR-GECO2 (plasmid no. 159603), pAAV-NIR-GECO2G (plasmid no.

## Abstract

Near-infrared (NIR) genetically encoded calcium ion ($Ca^{2+}$) indicators (GECIs) can provide advantages over visible wavelength fluorescent GECIs in terms of reduced phototoxicity, minimal spectral cross talk with visible light excitable optogenetic tools and fluorescent probes, and decreased scattering and absorption in mammalian tissues. Our previously reported NIR GECI, NIR-GECO1, has these advantages but also has several disadvantages including lower brightness and limited fluorescence response compared to state-of-the-art visible wavelength GECIs, when used for imaging of neuronal activity. Here, we report 2 improved NIR GECI variants, designated NIR-GECO2 and NIR-GECO2G, derived from NIR-GECO1. We characterized the performance of the new NIR GECIs in cultured cells, acute mouse brain slices, and *Caenorhabditis elegans* and *Xenopus laevis* in vivo. Our results demonstrate that NIR-GECO2 and NIR-GECO2G provide substantial improvements over NIR-GECO1 for imaging of neuronal $Ca^{2+}$ dynamics.

## Introduction

Fluorescence imaging of intracellular calcium ion ($Ca^{2+}$) transients using genetically encoded $Ca^{2+}$ indicators (GECIs) is a powerful and effective technique to monitor in vivo neuron activity in model organisms [1–4]. Over a time frame spanning 2 decades [5,6], tremendous effort has been invested in the development of visible wavelength GECIs based on green and red fluorescent proteins (GFP and RFP, respectively). These efforts have produced a series of high-performance GECIs that are highly optimized in terms of brightness, kinetics, $Ca^{2+}$ affinities,

159605), and pSF11-wNIR-GECO2-T2A-HO1 (plasmid no.159606) are available via Addgene according to the terms of the Uniform Biological Material Transfer Agreement.

**Funding:** Work in the lab of R.E.C. was supported by grants from CIHR (MOP-123514 and FS-154310), NSERC (RGPIN 288338-2010 and 2018-04364), Brain Canada, and NIH (U01 NS094246 and UO1 NS090565). E.S.B. acknowledges Lisa Yang, John Doerr, the HHMI-Simons Faculty Scholars Program, the HHMI Investigator Program, U.S.-Israel Binational Science Foundation (2014509), NIH (2R01-DA029639, 1R01-MH12297101, 1R01-DA045549, and 1RF1-NS113287), Human Frontier Science Program (RGP0015/2016), and NSF (1848029 and 1734870). Work in the lab of E.S.R. was supported by grants from CIHR (FDN-143238) and Brain Canada. Work in the lab of P.W.W. was supported by a grant from NSERC (RGPIN-2017-05005). K.D. P. acknowledges the Foundation of Westlake University. S.A. thanks Conrad F. Harrington Fellowship from the faculty of medicine at the University of McGill. The funders had no role in study design, data collection and analysis, decision to publish, or preparation of the manuscript.

**Competing interests:** The authors declare no competing interests.

**Abbreviations:** ACSF, artificial cerebrospinal fluid; AP, action potential; ATR, All-trans-retinal; BV, biliverdin; $Ca^{2+}$, calcium ion; CaM, calmodulin; ChR2, channelrhodopsin-2; DIV, days in vitro; EC, extinction coefficient; FBS, fetal bovine serum; GECI, genetically encoded calcium ion indicator; GFP, green fluorescent protein; HBSS, Hank's Balanced Salt Solution; HO1, heme oxygenase-1; iPSC-CM, induced pluripotent stem cell-derived cardiomyocyte; IUE, in utero electroporation; LB, Lysogeny broth; MEM, Minimum Essential Medium; NIR, near-infrared; QY, quantum yield; RFP, red fluorescent protein; SBR, signal-to-background ratio; SD, standard deviation; SEM, standard error of the mean; SNR, signal-to-noise ratio.

cooperativity, and resting (baseline) fluorescence [2–4,7]. In contrast, efforts to develop GECIs with near-infrared (NIR) excitation and emission (>650 nm) are at a relatively nascent state [8,9].

We recently described the conversion of an NIR FP (mIFP, a biliverdin (BV)-binding NIR FP engineered from the PAS and GAF domains of *Bradyrhizobium* bacteriophytochrome) [10] into a GECI designated NIR-GECO1. NIR-GECO1 was engineered by genetic insertion of the $Ca^{2+}$-responsive domain calmodulin (CaM)-RS20 into the protein loop close to the BV binding site of mIFP [8]. NIR-GECO1 provides a robust inverted fluorescence response (i.e., a fluorescence decrease upon $Ca^{2+}$ increase) in response to $Ca^{2+}$ concentration changes in cultured cells, primary neurons, and acute brain slices. Due to its NIR fluorescence, it has inherent advantages relative to visible wavelength GFP- and RFP-based GECIs including reduced phototoxicity, minimal spectral cross talk with visible light excitable optogenetic tools and fluorescent probes, and decreased scattering and absorption of excitation and emission light in mammalian tissues. However, as a first-generation GECI, NIR-GECO1 is suboptimal by several metrics including relatively low brightness and limited fluorescence response (i.e., $\Delta F/F_0$ for a given change in $Ca^{2+}$ concentration), which limit its utility for in vivo imaging of neuronal activity.

## Results

### Development of new NIR-GECO variants

Based on ample precedent from the development of the GFP-based GECI GCaMP series [1,3] and the RFP-based R-GECO series [2,11], we reasoned that NIR-GECO1 was likely to be amenable to further improvement by protein engineering and directed molecular evolution. Here, we report that such an effort has led to the development of 2 second-generation variants, designated NIR-GECO2 and NIR-GECO2G, which enable fluorescence imaging of neuronal activity–associated changes in intracellular $Ca^{2+}$ concentration with substantially greater sensitivity than their first-generation progenitor.

Starting from the template of the gene-encoding NIR-GECO1, 3 rounds of directed evolution were performed as described previously [8]. Briefly, the gene-encoding NIR-GECO1 was randomly mutated by error-prone PCR, and the resulting gene library was used to transform *Escherichia coli*. Petri dishes of colonies were screened using a fluorescence macro-imaging system, brightly fluorescent clones were picked and, following overnight culture, the NIR-GECO1 protein variants were extracted and tested for $Ca^{2+}$ responsiveness in vitro. The genes encoding the best variants, as determined by the in vitro test, were assessed for brightness and $Ca^{2+}$ responsiveness when expressed in HeLa cells. The gene that provided the best balance of brightness and function in HeLa cells was used as the template for a subsequent round of combined bacterial and HeLa cell screening. Three rounds of screening led to the identification of 2 promising variants: NIR-GECO2 (equivalent to NIR-GECO1 with T234I, S251T, E259G, Q402E, F463Y, and T478A, numbered as in S1 Fig) and NIR-GECO2G (equivalent to NIR-GECO2 with T251S and S347G; Fig 1A). In terms of fluorescence spectral profile, peak maxima, extinction coefficient (EC), quantum yield (QY), and $pK_a$, NIR-GECO2 and NIR-GECO2G are essentially identical to NIR-GECO1 (S1 Table). One of the most pronounced changes in the biophysical properties is that the $Ca^{2+}$ affinities of NIR-GECO2 and NIR-GECO2G are higher than that of NIR-GECO1 with $K_d$ values of 331 nM and 480 nM, respectively ($K_d$ of NIR-GECO1 is 885 nM) (S2A Fig). A parallel effort to construct a second-generation NIR-GECO1 by replacing the mIFP portion with the brighter and homologous miRFP [12] resulted in the functional indicator prototype. However, further optimization was abandoned due to the apparent toxicity of the miRFP-based construct when expressed in *E. coli* (S3 and S4 Figs).

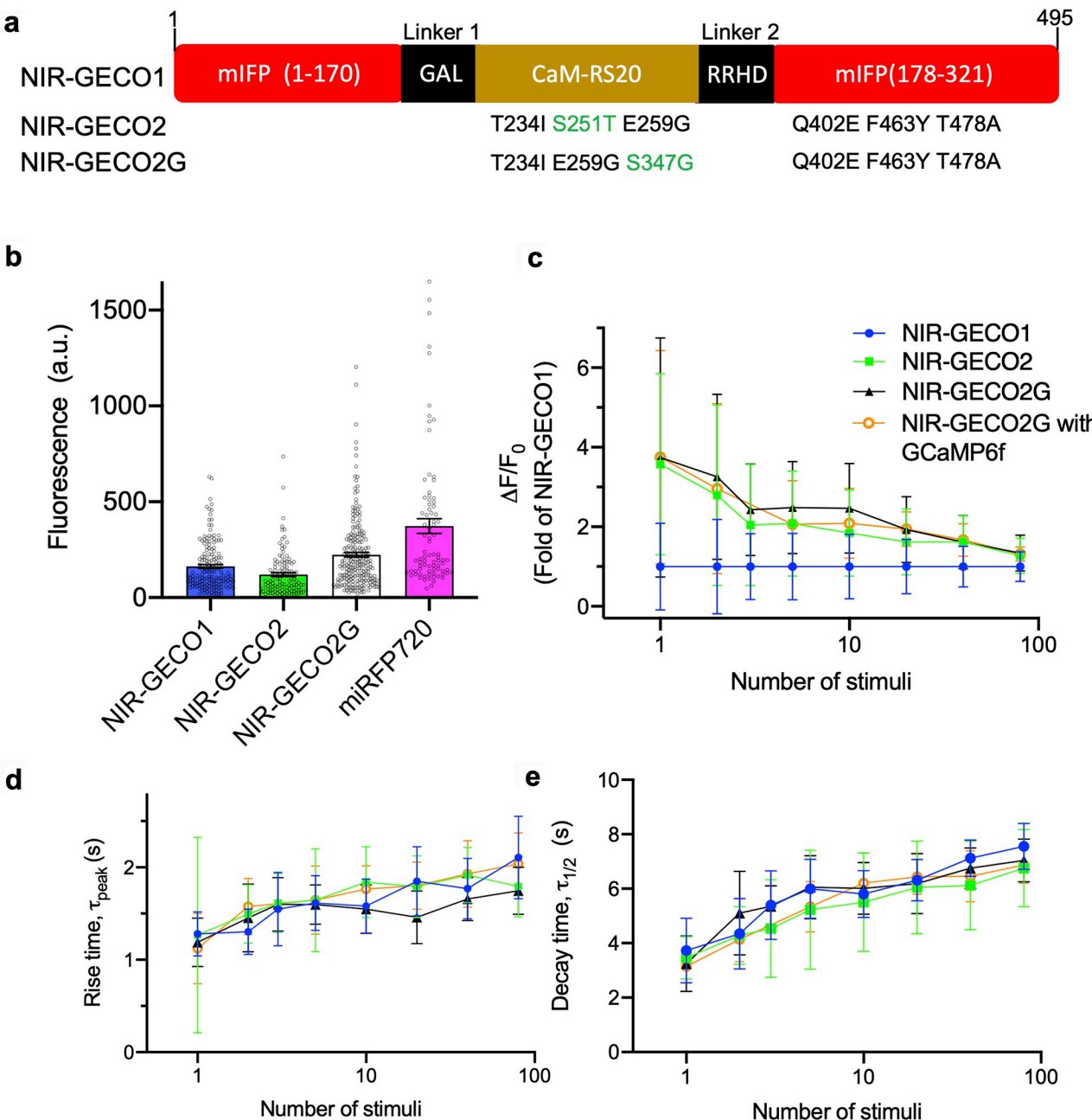

**Fig 1. NIR-GECO evolution and characterization in dissociated neurons.** (**a**) Mutations of NIR-GECO2 and NIR-GECO2G relative to NIRGECO1. The different mutations between NIR-GECO2 and NIR-GECO2G are highlighted in green. (**b**) Relative fluorescence intensity (mean ± SEM) of NIR-GECO1, NIR-GECO2, NIR-GECO2G, and miRFP720 in neurons ($n$ = 160, 120, 219, and 84 neurons, respectively, from 2 cultures). Fluorescence was normalized by co-expression of GFP via self-cleavable 2A peptide. (**c–e**) Comparison of NIR-GECO variants, as a function of stimulus strength (the same color code is used in panels **c–e**). (**c**) $\Delta F/F_0$; (**d**) rise time; (**e**) half decay time. Values are shown as mean ± SD ($n$ = 10 wells from 3 cultures). The underlying data for (**b**) to (**e**) can be found in S1 Data. GFP, green fluorescent protein; NIR, near-infrared; SD, standard deviation; SEM, standard error of the mean.

## Characterization of new NIR-GECO variants

To compare the intracellular baseline brightness (i.e., fluorescence in a resting state neuron) of NIR-GECO2 and NIR-GECO2G with the previously reported NIR-GECO1 and the NIR FP

miRFP720 [13], we expressed each construct in cultured neurons and quantified the overall cellular brightness 5 days after transfection. To correct for cell-to-cell variations in protein expression, we co-expressed GFP stoichiometrically via a self-cleaving 2A peptide [14] to serve as an internal reference for expression level. Under these conditions, NIR-GECO2 is approximately 25% dimmer than NIR-GECO1 while NIR-GECO2G is 50% brighter than NIR-GECO1 (Fig 1B). Photobleaching experiments for the neuronally expressed constructs revealed that all the NIR-GECO variants possess similar photostability and photobleach approximately 4× faster than miRFP720 (S2B Fig). Under an excitation intensity of 2.8 mW/mm$^2$ and an exposure time of 100 ms, which was generally used for wide-field imaging of NIR-GECO2 and NIR-GECO2G, NIR-GECO2 retained 89.4%, 67.8%, 57.1%, and 48.7% of original fluorescence intensity following 30 minutes of imaging at 1 Hz, 2 Hz, 5 Hz, and continuous illumination, respectively. Under the same imaging conditions, NIR-GECO2G had 94.1%, 85.3%, 68.0%, and 49.5% of the initial brightness remaining after 30 minutes of imaging at 1 Hz, 2 Hz, 5 Hz, and continuous illumination, respectively (S2C and S2D Fig). Based on these results, we recommend 2 Hz (with 100-ms exposure time) as the maximum acquisition rate for imaging of NIR-GECO2 and NIR-GECO2G.

To assess the sensitivity (i.e., $-\Delta F/F_0$) of the new NIR-GECO variants, electric field stimulation [15] was performed to cultured neurons expressing NIR-GECO2, NIR-GECO2G, and NIR-GECO1. For a single field stimulation-evoked action potential (AP), NIR-GECO2 and NIR-GECO2G responded with similar $-\Delta F/F_0$ values of 16% and 17%, respectively. These values are 3.6- and 3.7-fold higher, respectively, than that of NIR-GECO1 (4.5%). For small numbers of APs (2 to 10), the responses of NIR-GECO2G were 2.5- to 3.3-fold larger than those of NIR-GECO1 and the responses of NIR-GECO2 were 1.8- to 2.8-fold larger than those of NIR-GECO1. At higher numbers of APs (20 to 80), the improvements of the new variants became less pronounced as the $\Delta F/F_0$ values of the 3 variants converged (Fig 1C). The on (rise time, $\tau_{peak}$; Fig 1D) and off (decay time, $\tau_{1/2}$; Fig 1E) kinetics of NIR-GECO2 and NIR-GECO2G, in response to field stimulation-evoked APs stimuli, remained similar to that of NIR-GECO1. We also characterized NIR-GECO2G with co-expression of GCaMP6f to see whether the presence of GCaMP6f has effects on the performance of NIR-GECO2G. The brightness of NIR-GECO2G decreased by approximately 16% when co-expressed with GCaMP6f (S2E Fig), but other key properties (i.e., $\Delta F/F_0$ and on- and off-kinetics) were effectively unchanged (Fig 1C–1E).

To investigate if NIR-GECO2 and NIR-GECO2G provide advantages over NIR-GECO1 for combined use with an optogenetic actuator, we co-transfected HeLa cells with the genes encoding Opto-CRAC and each of the 3 NIR-GECO variants. Opto-CRAC is an optogenetic tool that can be used to induce $Ca^{2+}$ influx into non-excitatory cells when illuminated with blue light [16]. Transfected HeLa cells were illuminated with 470-nm light at a power of 1.9 mW/mm$^2$, while the NIR fluorescence intensity of NIR-GECO variants was continuously recorded. Following 100 ms of blue light stimulation, the average $-\Delta F/F_0$ for NIR-GECO2G, NIR-GECO2, and NIR-GECO1 was 34.5%, 22.8%, and 12.1%, respectively (Fig 2A and 2D). With 500 ms of illumination, the $-\Delta F/F_0$ values increased to 48.2%, 42.1%, and 30.7%, respectively. At 1 second of illumination time, the $-\Delta F/F_0$ was 40.7%, 38.0%, and 33.3%, respectively (Fig 2A–2C). As a control, we also illuminated HeLa cells expressing only NIR-GECO2 or NIR-GECO2G with blue light (470 nm) at a 3.3-fold higher power intensity than that used in Fig 2 and found no artifactual fluorescence changes (S5 Fig). When expressed in acute brain slices, both NIR-GECO2 and NIR-GECO2G robustly reported $Ca^{2+}$ changes in neurons in response to either optogenetic (CoChR) or chemical (4-aminopyridine) stimulation (S6 Fig). These results support the conclusion that both NIR-GECO2G and NIR-GECO2 are more sensitive than NIR-GECO1 for reporting $Ca^{2+}$ transients at low $Ca^{2+}$ concentrations, and

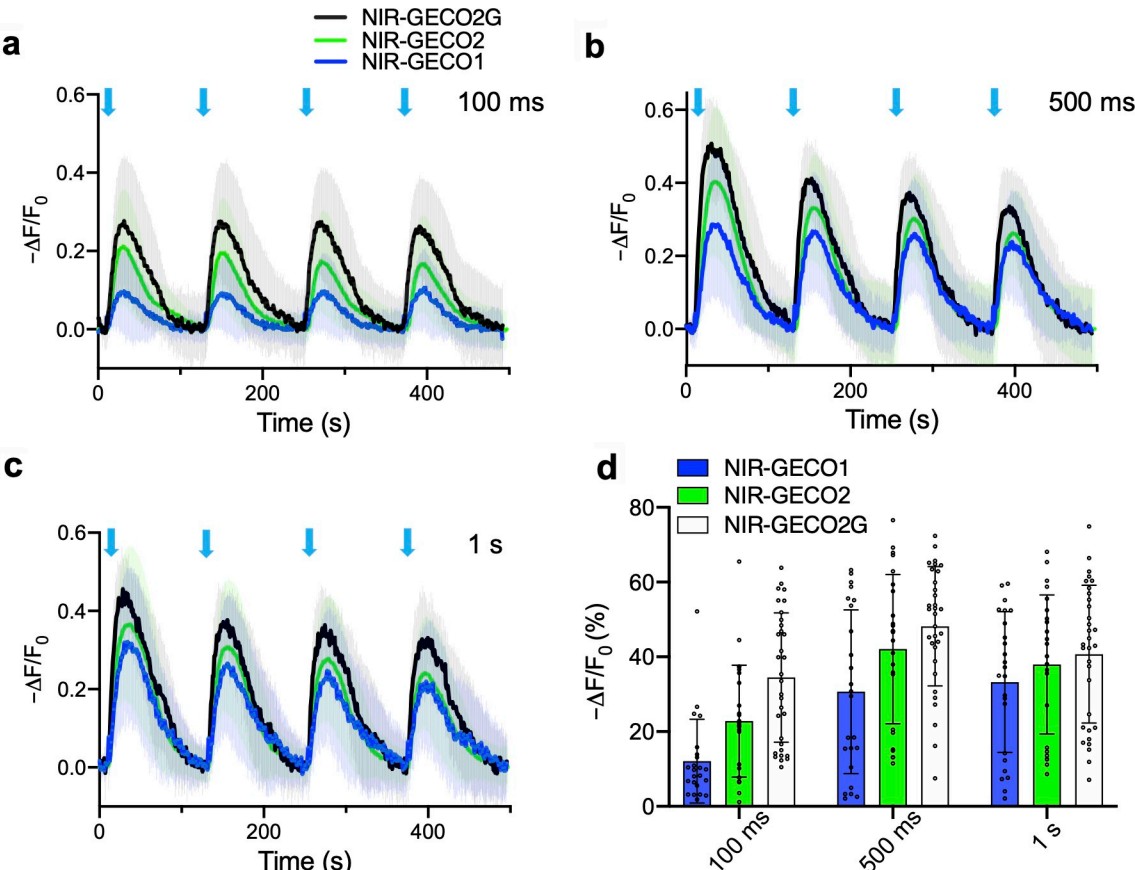

**Fig 2. Performance of NIR-GECO2G, NIR-GECO2, and NIR-GECO1 in HeLa cells.** (a–c) Fluorescence traces of NIR-GECO2G, NIR-GECO2, and NIR-GECO1 in response to 100 ms (**a**), 500 ms (**b**), and 1 s (**c**) blue light activation (470 nm at a power of 1.9 mW/mm$^2$) in HeLa cells with co-expression of Opto-CRAC. Opto-CRAC is composed of the STIM1-CT and LOV2 domain. The fusion of STIM1-CT to the LOV2 domain allows photo-controllable exposure of the active site of SRIMI-CT, which is able to interact with ORAI1 and trigger Ca$^{2+}$ entries across the plasma membrane [16]. Black, green, and dark blue lines represent averaged data for NIR-GECO2G ($n = 32$ cells), NIR-GECO2 ($n = 25$ cells), and NIR-GECO1 ($n = 23$ cells), respectively. The same color code is used in panels **a–c**. Shaded areas represent the SD. (**d**) Quantitative -ΔF/F$_0$ (mean ± SD) for NIR-GECO2G, NIR-GECO2, and NIR-GECO1 in **a–c**. The underlying data for **a–d** can be found in S1 Data. NIR, near-infrared; SD, standard deviation.

NIR-GECO2G is the best of the 3. To further explore the use of NIR-GECO2G, we expressed it in human induced pluripotent stem cell-derived cardiomyocytes (iPSC-CMs). In iPSC-CMs, NIR-GECO2G enabled robust imaging of spontaneous Ca$^{2+}$ oscillations, caffeine induced Ca$^{2+}$ influx, and channelrhodopsin-2 (ChR2)-dependent activation (S7 Fig).

## In vivo imaging of Ca$^{2+}$ in *C. elegans* using NIR-GECO2

To determine if NIR-GECO2 was suitable for in vivo imaging of neuronal activity, we first sought to test it in *Caenorhabditis elegans*, a popular model organism in neuroscience. For this application, we chose to use NIR-GECO2 rather than NIR-GECO2G due to its higher Ca$^{2+}$ affinity; however, NIR-GECO2G could be also readily expressed in *C. elegans* in neurons producing sufficient NIR fluorescence (S8d Fig). As the internal BV concentration of *C. elegans* is quite low due to its inability to synthesize heme de novo (its main source of heme is from the ingestion of *E. coli*) [17,18], *C. elegans* expressing NIR-GECO2 pan-neuronally showed no detectable NIR fluorescence in the brain region (S9 Fig). Thus, we decided to co-express heme oxygenase-1 (HO1) to increase the conversion of heme into BV [19]. We created *C. elegans*

lines expressing NLS-NIR-GECO2-T2A-HO1 (where NLS is a nuclear localization sequence) and NLS-jGCaMP7s under the pan-neuronal tag-168 promoter in an extrachromosomal array. The resulting transgenic worms exhibited bright nuclear localized fluorescence from both NIR-GECO2 and jGCaMP7s (Fig 3A). One notable advantage of the NIR-GECO series relative to the GCaMP series of indicators is the lower autofluorescence in the intestinal area of worms (Fig 3A, S8A and S8D Fig). In worms expressing both NLS-jGCaMP7s and NIR-GECO2, the resting state cellular brightness of NIR-GECO2 was approximately 1.5-fold higher than that of jGCaMP7s (Fig 4A), and the signal-to-background ratio (SBR; i.e., the ratio of fluorescence emitted from neurons to autofluorescence from the intestine area) of NIR-GECO2 was approximately 5-fold larger than that of jGCaMP7s (Fig 4B). The ratio of SBRs for NIR-GECO2 versus jGCaMP7s at different depths into worms (down to the bottom of the worm at approximately 20 μm) revealed no substantial differences as a function of depth (Fig 4D).

For functional imaging of NIR-GECO2, microfluidic chips [20] were used to deliver a high osmotic strength stimulus (200 mM NaCl) to individual worms, and the fluorescence was imaged simultaneously in the NIR and green fluorescence channels. Following exposure to a high concentration of NaCl, we detected synchronous but opposing fluorescent changes for jGCaMP7s (fluorescence increases) and NIR-GECO2 (fluorescence decreases) (Fig 3B). Quantitative analysis of 36 spikes from 3 neurons showed that the $-\Delta F/F_0$ of NIR-GECO2 was about half of the $\Delta F/F_0$ of jGCaMP7s following NaCl stimulation ($\Delta F/F_0 = 0.39 \pm 0.19$ for jGCaMP7s; $-\Delta F/F_0 = 0.19 \pm 0.07$ for NIR-GECO2; Fig 3C). We also quantified signal-to-noise ratio (SNR) of jGCaMP7s and NIR-GECO2 from spontaneously firing neurons in *C. elegans*. NIR-GECO2 and jGCaMP7s exhibited similar SNRs (Fig 4C) and neither GECI showed a substantial advantage as a function of imaging depth, consistent with what we found with SBRs (Fig 4E).

We next attempted all-optical stimulation and imaging of neuron activity in *C. elegans* using the blue light-sensitive channelrhodopsin CoChR [21] and NIR-GECO2. We previously demonstrated that excitation wavelengths used to image NIR-GECO1 do not activate CoChR [8]. NIR-GECO2 (with co-expression of HO1) was expressed in AVA interneurons (involved in backward locomotion) under the flp-18 promoter, and CoChR-GFP was expressed in upstream ASH neurons under control of the sra-6 promoter. Imaging of transgenic worms with confocal microscopy revealed 2 AVA neurons with expression of NIR-GECO2 and 2 ASH neurons with expression of CoChR (Fig 3D). Blue light stimulation of CoChR in ASH neurons caused long-lasting (tens of seconds to a few minutes) fluorescent decreases in NIR-GECO2 fluorescence ($-\Delta F/F_0$ of 30% to 90%) in the downstream AVA interneurons (Fig 3E). Collectively, these data indicate that the combination of NIR-GECO2 and CoChR provides a robust all-optical method to interrogate hierarchical circuits in *C. elegans*.

## In vivo imaging of Ca²⁺ in *Xenopus laevis* using NIR-GECO2G

To further evaluate the utility of NIR-GECO2G for in vivo imaging of neuronal activity in a vertebrate brain, we transiently expressed the genes encoding NIR-GECO2G (without co-expression of HO1) and GCaMP6s [1] in *Xenopus laevis* tadpoles by mRNA injection into early embryos. Light-sheet microscopy imaging of the olfactory bulb of live tadpoles revealed that individual neurons exhibited strong NIR fluorescent signals due to NIR-GECO2G expression (Fig 5A and 5B, S1 Video). Fluorescence from both NIR-GECO2G and GCaMP6s was observed to oscillate in a synchronous but opposing manner in response to spontaneous neuronal activity (Fig 5C). These results demonstrate that NIR-GECO2G can be used to report dynamic Ca²⁺ changes in vivo in *X. laevis* in the absence of added BV or HO1 co-expression.

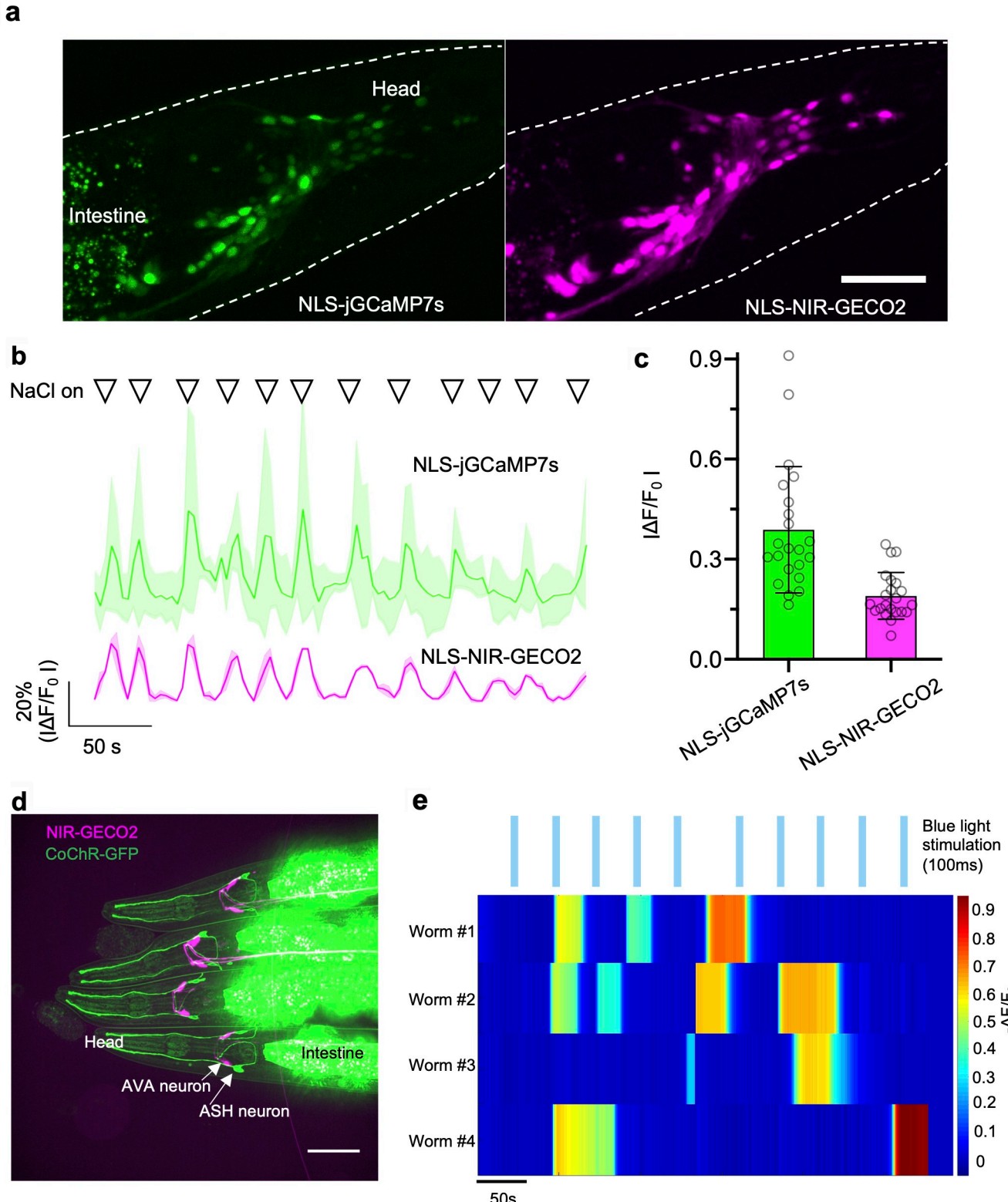

**Fig 3. Imaging of NIR-GECO2 in response to microfluidic and optogenetic stimulation in *C. elegans* in vivo.** (**a**) Left, fluorescent image of neurons expressing NLS-jGCaMP7s ($\lambda_{ex}$ = 488-nm laser light, $\lambda_{em}$ = 527/50 nm). Right, fluorescent image of neurons expressing NIR-GECO2-T2A-HO1 ($\lambda_{ex}$ = 640-nm laser light, $\lambda_{em}$ = 685/40 nm). Representative of more than 3 worms, both under tag-168 promoter. Scale bar, 50 μm. (**b**) Fluorescence traces of NLS-jGCaMP7s (top) and NLS-NIR-GECO2 (bottom) in response to the stimulation of microfluidic containing 200 mM NaCl. Solid lines represent averaged

data from 3 neurons. Shaded areas are shown as SD. Triangles on the top of the traces indicate the time points of stimulation (20 seconds for each stimulation). (**c**) Quantitative fluorescence changes of NLS-jGCaMP7s and NLS-NIR-GECO2 in **b** ($n$ = 36 spikes from 3 neurons). (**d**) Fluorescence image of the 4 *C. elegans* expressing NIR-GECO2-T2A-HO1 in AVA neurons (under flp-18 promoter) and CoChR-GFP in ASH neurons (under sra-6 promoter). The merged image is shown. Imaging conditions: NIR-GECO2, $\lambda_{ex}$ = 640-nm laser light, $\lambda_{em}$ = 685/40; GFP, $\lambda_{ex}$ = 488-nm laser light, $\lambda_{ex}$ = 527/50 nm. (**e**) Individual traces of NIR-GECO2 fluorescence in an AVA neuron under blue light illumination (20 mW/mm$^2$, $\lambda_{ex}$ = 488-nm laser light, 100 ms; blue bars). The underlying data for **b**, **c**, and **e** can be found in S1 Data. NIR, near-infrared; SD, standard deviation.

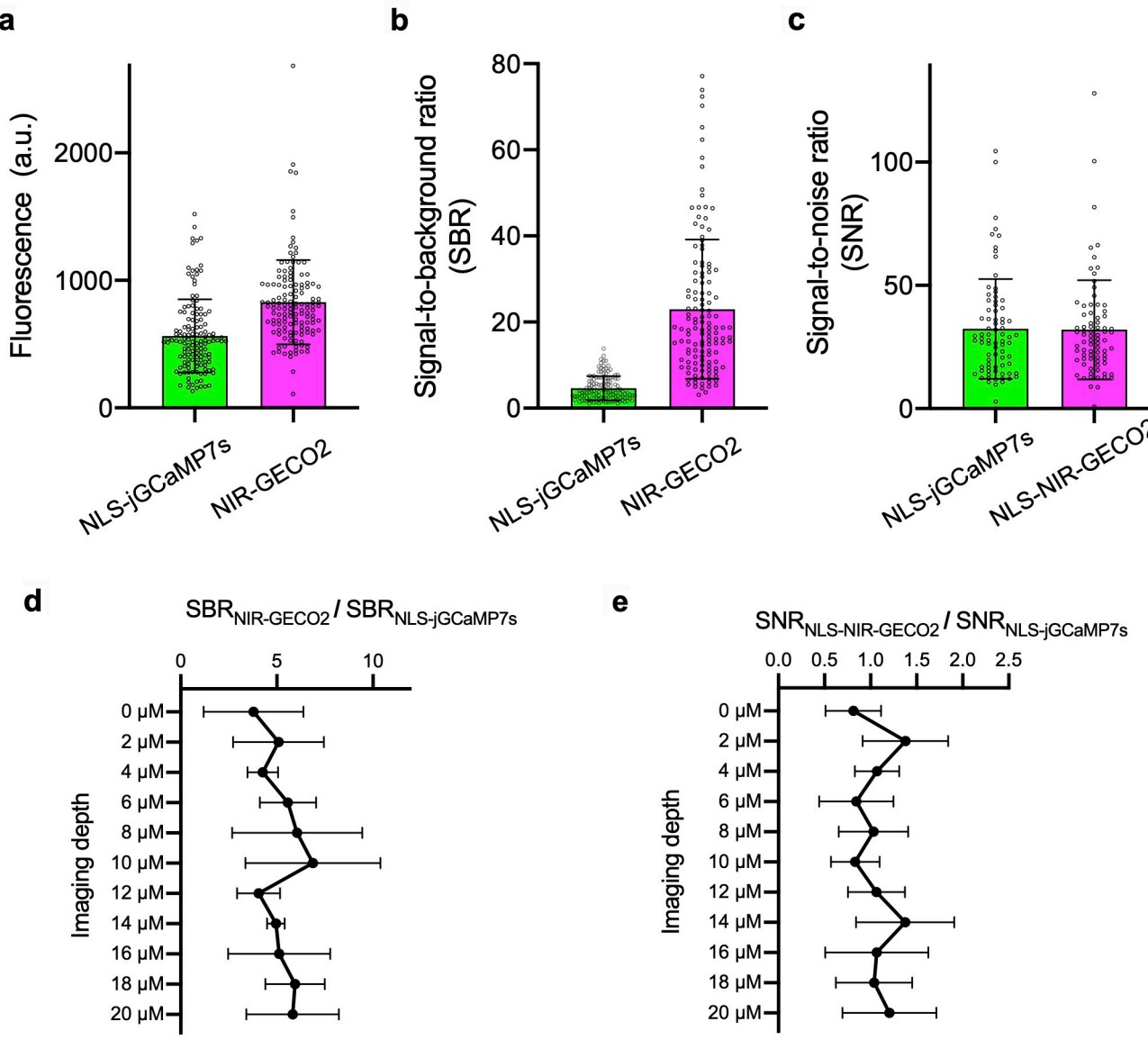

**Fig 4. Comparison of jGCaMP7s and NIR-GECO2 in *C. elegans*.** (**a**) Fluorescence intensity of NLS-jGCaMP7s and NIR-GECO2 in *C. elegans* neurons at resting state. Fluorescence was normalized to the same excitation intensity ($n$ = 132 ROIs from 5 worms; data are shown as mean ± SD) (**b**) SBR of NLS-jGCaMP7s and NIR-GECO2 in neurons of *C. elegans* at resting state ($n$ = 132 ROIs from 5 worms; data are shown as mean ± SD). SBR was obtained via dividing the fluorescence intensity from neurons by the averaged autofluorescence from the intestine area. (**c**) SNR of NLS-jGCaMP7s and NLS-NIR-GECO2 quantified from spontaneously spiking neurons ($n$ = 78 ROIs from 4 worms; data are shown as mean ± SD). SNR was calculated by dividing the fluorescence change associated with a spike by the SD of the baseline fluorescence over the 2-second period immediately before the spike. (**d**) The ratio of SBR$_{NIR-GECO2}$ to SBR$_{NLS-jGCaMP7s}$ at different imaging depths ($n$ = 5 worms; data are shown as mean ± SD). (**e**) The ratio of SNR$_{NLS-NIR-GECO2}$ to SNR$_{NLS-jGCaMP7s}$ at different imaging depths ($n$ = 4 worms; data are shown as mean ± SD). NIR-GECO2 (without NLS) and NLS-jGCaMP7s were used for the experiments in **a**, **b**, and **d**; NLS-NIR-GECO2 and NLS-jGCaMP7s were used for the experiments in **c** and **e**. The underlying data for **a**–**e** can be found in S1 Data. ROI, region of interest; SBR, signal-to-background ratio; SD, standard deviation; SNR, signal-to-noise ratio.

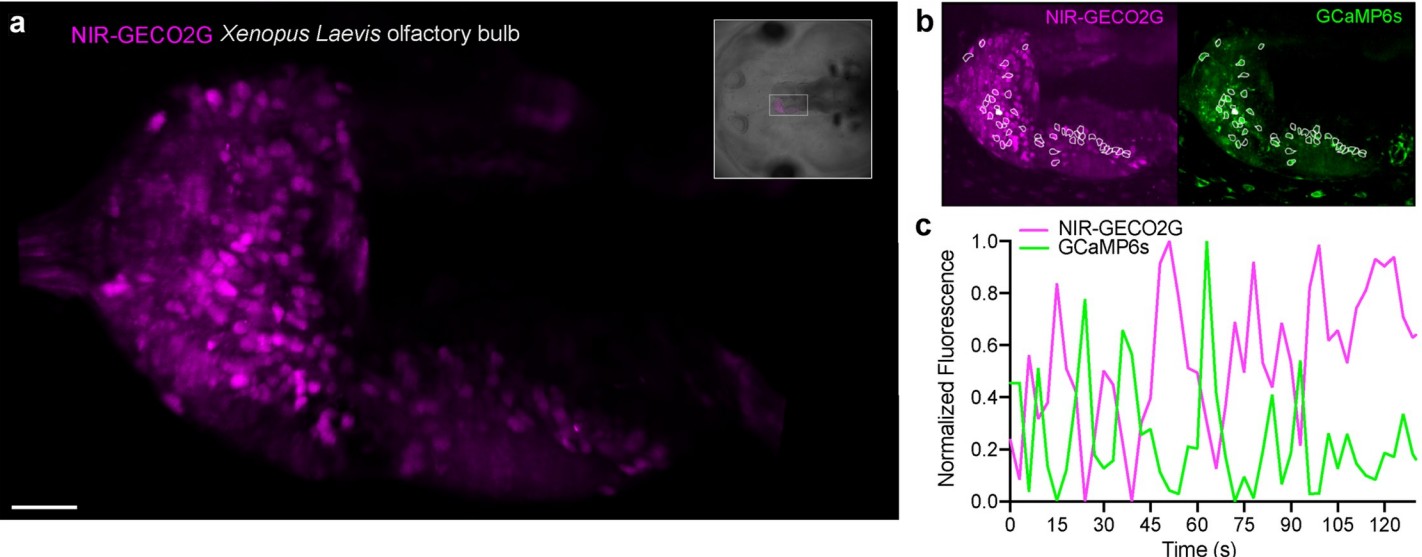

**Fig 5. Spontaneous Ca²⁺ response imaging with NIR-GECO2G in the olfactory bulb of *Xenopus laevis*.** (**a**) Light-sheet acquisition of spontaneous Ca²⁺ response in the olfactory bulb in an intact animal. The image shows an intensity projection over 200 frames and a volume spanning 45 μm. The orientation and position of the field of view is marked in the brightfield image of the animal (upper right frame). Scale bar, 40 μm. (**b**) White frames indicate the cells that showed Ca²⁺ response with the NIR-GECO2G (magenta) and GCaMP6s (green) indicator. (**c**) A representative fluorescence response trace of spontaneous activity of 1 cell in the olfactory bulb (solid white ROI in **b**) is plotted over time. The intensity responses of NIR-GECO2G and GCaMP6s are antagonal to each other. Z-series volumes were acquired once every 3.0 seconds. NIR, near-infrared; ROI, region of interest.

## Discussion

In summary, we have developed 2 improved NIR fluorescent Ca²⁺ indicators designated NIR-GECO2 and NIR-GECO2G. Of the 2, NIR-GECO2 has higher response amplitudes but dimmer fluorescence compared to NIR-GECO1, based on characterization in neurons and HeLa cells. In contrast, NIR-GECO2G is improved relative to NIR-GECO1 in terms of both overall cellular brightness (approximately 50% brighter than NIR-GECO1) and sensitivity (up to an approximately 3.7-fold improvement in $-\Delta F/F_0$ relative to NIR-GECO1 for single AP). As we have demonstrated in this work, these improvements make the new variants particularly useful for imaging Ca²⁺ dynamics in small model organisms. Specifically, NIR-GECO2 offers comparable sensitivity to jGCaMP7s in *C. elegans* and NIR-GECO2G enables robust imaging of Ca²⁺ dynamics in the olfactory bulb of *X. laevis* tadpoles. With their NIR excitation and emission, improved sensitivity, and ability to be subcellularly targeted (e.g., with an NLS), NIR-GECO2 and NIR-GECO2G should prove useful for multicolor and multi-compartment imaging when combined with other fluorescent probes.

However, even with the improvements described in this work, NIR GECIs still face challenges including lower brightness, slower kinetics, and faster photobleaching compared to the state-of-art green and red fluorescent GECIs. For these reasons, it remains challenging to use NIR-GECO2 and NIR-GECO2G to image Ca²⁺ dynamics with single-cell resolution in rodents where neuronal BV concentrations are low and cannot be substantially increased by the strategy of co-expressing HO1 [8]. A promising approach to overcome this challenge was recently described by Kobachi and colleagues, who demonstrated that knocking out the gene for BV reductase increases the BV concentration in mice and improves the brightness of NIR-GECO1 by 4.3-fold [22]. While the off-kinetics of NIR-GECOs are very similar to that of GCaMP6s or jGCaMP7s, the on-kinetics of NIR-GECO series are slower than that of GCaMP series. These slower on-kinetics may mean that closely spaced Ca²⁺ peaks, which could be

resolved with a GCaMP variant, will appear to merge into a single peak with NIR-GECO variants. Finally, photobleaching of NIR-GECO2(G) was not a major limitation for the 1-photon imaging experiments reported here, but may be a concern for long duration 1-photon imaging experiments or other types of imaging techniques (such as photoacoustic imaging) where strong illumination power is required. Overcoming these challenges will undoubtedly require further directed molecular evolution and optimization of the NIR-GECO series or the possible development of alternative NIR GECI designs based on brighter and more photostable NIR FP scaffolds.

## Materials and methods

### Mutagenesis and molecular cloning

Synthetic DNA oligonucleotides used for cloning and library construction were purchased from Integrated DNA Technologies (Coralville, Iowa, United States of America). Random mutagenesis of NIR-GECO variants was performed using *Taq* DNA polymerase (New England Biolabs, Ipswich, Massachusetts, USA) with conditions that resulted in a mutation frequency of 1 to 2 mutations per 1,000 base pairs. Gene fragments for NIR-GECO libraries were then inserted between restriction sites *XhoI* and *HindIII* of pcDuex2 for expression. The DNA sequences encoding $miRFP_{1\ to\ 172}$, CaM-RS20 (from NIR-GECO1), and $miRFP_{179\ to\ 311}$ were amplified by PCR amplification separately and then used as DNA templates for the assembly of miRFP1 to 172—CaM-RS20—$miRFP_{179\ to\ 311}$ by overlap extension PCR. The resulting DNA sequence was then digested and ligated into the pcDNA3.1(-) vector for mammalian expression and into a pBAD-MychisC (Invitrogen, Waltham, Massachusetts, USA) vector for bacterial expression. Q5 high-fidelity DNA polymerase (New England Biolabs) was used for routine PCR amplification and overlap extension PCR. PCR products and products of restriction digests were routinely purified using preparative agarose gel electrophoresis followed by DNA isolation with the GeneJET gel extraction kit (Thermo Fisher Scientific, Waltham, Massachusetts, USA). Restriction endonucleases were purchased from Thermo Fisher Scientific. Ligations were performed using T4 ligase in Rapid Ligation Buffer (Thermo Fisher Scientific).

Genes or gene libraries in expression plasmids were electroporated into *E. coli* strain DH10B (Thermo Fisher Scientific). The transformed cells were then plated on 10-cm Lysogeny broth (LB) agar Petri dishes supplemented with 400 μg/mL ampicillin (Sigma, St. Louis, Missouri, USA) and 0.0004% (wt/vol) L-arabinose (Alfa Aesar, Haverhill, Massachusetts, USA) at 37°C overnight. For library screening, bright bacterial colonies expressing NIR-GECO variants were picked and cultured. Proteins were extracted using B-PER (Thermo Fisher Scientific) from overnight cultures of bacteria growing in LB media supplemented with 100 μg/mL ampicillin and 0.0016% L-arabinose and then tested for fluorescence and $Ca^{2+}$ response in 384-well plates. Variants with the highest brightness and $Ca^{2+}$ response were selected, and the corresponding plasmids were purified. HeLa cells were transfected with the selected plasmids, and live-cell fluorescence imaging was used to reevaluate both brightness and $Ca^{2+}$ response. Small-scale isolation of plasmid DNA was done with GeneJET miniprep kit (Thermo Fisher Scientific).

For the construction of Opto-CRAC-EYFP, a synthetic double-stranded DNA fragment consisting of fused EYFP, LOV2, and STIM1-CT fragments (residues 336–486) [16], flanked with *NotI* and *XhoI* restriction sites, was cloned into the pcDNA3.1(-) vector.

### Protein purification and in vitro characterization

The genes for the miRFP-based $Ca^{2+}$ indicators NIR-GECO1, NIR-GECO2, and NIR-GECO2G, with a poly-histidine tag on the C-terminus, were expressed from a pBAD-MychisC

(Invitrogen) vector containing the gene of cyanobacteria *Synechocystis* HO1 as previously described [23,24]. Bacteria were lysed with a cell disruptor (Constant Systems, Northants, United Kingdom) and then centrifuged at 15,000$g$ for 30 minutes, and proteins were purified by Ni-NTA affinity chromatography (Agarose Bead Technologies, Doral, Florida, USA). The buffer was exchanged to 10 mM MOPS, 100 mM KCl (pH 7.2) with centrifugal concentrators (GE Healthcare Life Sciences, Vancouver, British Columbia, Canada). The spectra of miRFP-based $Ca^{2+}$ indicator prototype, with and without $Ca^{2+}$, were measured in a 384-well plate. Briefly, purified proteins were loaded into 384-well plates and then supplied with either 10 mM EGTA or 5 mM CaCl$_2$ before measuring emission spectra. The ECs, QY, and p$K_a$ for NIR-GECO variants were determined as previously described [8]. $Ca^{2+}$ titrations of NIR--GECO variants were performed with EGTA-buffered $Ca^{2+}$ solutions. We prepared buffers by mixing a CaEGTA buffer (30 mM MOPS, 100 mM KCl, 10 mM EGTA, and 10 mM CaCl$_2$) and an EGTA buffer (30 mM MOPS, 100 mM KCl, and 10 mM EGTA) to give free $Ca^{2+}$ concentrations ranging from 0 nM to 39 μM at 25˚C. Fluorescence intensities were plotted against $Ca^{2+}$ concentrations and fitted by a sigmoidal binding function to determine the Hill coefficient and $K_d$. To determine k$_{off}$ for NIR-GECO variants, an SX20 stopped-flow spectrometer (Applied Photophysics, Surrey, UK) was used. Proteins samples with 10-μM CaCl$_2$ (30 mM MOPS, 100 mM KCl, and pH 7.2) were rapidly mixed with 10 mM EGTA (30 mM MOPS, 100 mM KCl, and pH 7.2) at room temperature, and fluorescence growth curve was measured and fitted by a single exponential equation.

## Imaging of miRFP-based $Ca^{2+}$ indicator prototype in HeLa cells

HeLa cells (40% to 60% confluent) on 24-well glass bottom plate (Cellvis, Mountain View, California, USA) were transfected with 0.5 μg of plasmid DNA and 2 μl TurboFect (Thermo Fisher Scientific). Following 2-hour incubation, the media was changed to DEME (Gibco, Waltham, Massachusetts, USA) with 10% fetal bovine serum (FBS) (Sigma), 2 mM GlutaMax (Thermo Fisher Scientific), and 1% penicillin-streptomycin (Gibco), and the cells were incubated for 48 hours at 37˚C in a CO$_2$ incubator before imaging. Prior to imaging, culture medium was changed to Hank's Balanced Salt Solution (HBSS). Wide-field imaging was performed on a Nikon Eclipse Ti microscope that was equipped with a 75 W Nikon xenon lamp, a 16-bit 512SC QuantEM EMCCD (Photometrics, Tucson, Arizona, USA), and a 60× objective and was driven by a NIS-Elements AR 4.20 software package (Nikon, Tokyo, Japan). For time-lapse imaging, HeLa cells were treated with 4 mM EGTA (with 5-μM ionomycin) and then 10 mM CaCl$_2$ (with 5-μM ionomycin). Images were taken every 5 seconds using a filter set with 650/60-nm excitation and 720/60-nm emission.

## Imaging of NIR-GECO1 and NIR-GECO2G in HeLa cells with Opto-CRAC

HeLa cells were co-transfected with pcDNA3.1-NIR-GECO1 or pcDNA3.1-NIR-GECO2 and pcDNA3.1-Opto-CRAC-EYFP using transfection reagent Lipofectamine 3000 (Invitrogen) following the manufacturer's instructions. An inverted microscope (D1, Zeiss, Oberkochen, Germany) equipped with a 63× objective lens (NA 1.4, Zeiss) and a multiwavelength LED light source (pE-4000, CoolLED, Andover, UK) was used. Blue (470 nm) and red (635 nm) excitations were used to illuminate Opto-CRAC-EYFP and image NIR-GECO variants, respectively. The GFP filter set (BP 470–490, T495lpxr dichroic mirror, and HQ525/50 emission filter) and the NIR filter set (ET 650/45x, T685lpxr dichroic mirror, and ET720/60 emission filter) were used to confirm the expression of Opto-CRAC-EYFP and NIR-GECO variants. The filter set (T685lpxr dichroic mirror, and ET720/60 emission filter) was used to stimulate Opto-CRAC and to acquire fluorescence imaging of NIR-GECO1 and NIR-GECO2G. Optical stimulation

was achieved with the 470-nm LED light at a power density of 1.9 mW/mm$^2$. Photoconversion testing of NIR-GECO2 and NIR-GECO2G was performed with 470-nm LED light at a power density of 6.2 mW/mm$^2$. Fluorescence signals were recorded by a sCMOS camera (ORCA--Flash4.0LT, Hamamatsu, Hamamatsu City, Japan) and controlled by a software (HC Image or NIS-Elements Advanced Research).

## Imaging of NIR-GECO2G in Human iPSC-derived cardiomyocytes

Human iPSC-derived cardiomyocytes (human iPSC cardiomyocytes—male | ax2505) were purchased from Axol Bioscience (Cambridge, UK). The 96-well glass bottom plate was first coated with fibronectin and gelatin (0.5% and 0.1%, respectively) at 37˚C for at least 1 hour. The cells were then plated and cultured for 3 days in Axol's Cardiomyocyte Maintenance Medium. IPSC-CMs were then transfected with pcDNA3.1-NIR-GECO2 with or without pcDNA3.1-hChR2-EYFP using Lipofectamine 3000 (Invitrogen) following the manufacturer's instructions. The medium was switched to Tyrode's buffer right before imaging. Imaging was performed with an inverted microscope (D1, Zeiss) equipped with a 63× objective lens (NA 1.4, Zeiss) and a multiwavelength LED light source (pE-4000, CoolLED) using the same settings described above.

## Imaging of NIR-GECO1, NIR-GECO2, NIR-GECO2G, and miRFP720 in cultured neurons

For dissociated hippocampal mouse neuron culture preparation, postnatal day 0 or 1 Swiss Webster mice (Taconic Biosciences, Albany, New York, USA) were used as previously described [21]. Briefly, dissected hippocampal tissue was digested with 50 units of papain (Worthington Biochem, Lakewood, New Jersey, USA) for 6 to 8 minutes at 37˚C, and the digestion was stopped by incubating with ovomucoid trypsin inhibitor (Worthington Biochem) for 4 minutes at 37˚C. Tissue was then gently dissociated with Pasteur pipettes, and dissociated neurons were plated at a density of 20,000 to 30,000 per glass coverslip coated with Matrigel (BD Biosciences, San Jose, California, USA). Neurons were seeded in 100 μL plating medium containing Minimum Essential Medium (MEM) (Thermo Fisher Scientific), glucose (33 mM, Sigma), transferrin (0.01%, Sigma), HEPES (10 mM, Sigma), Glutagro (2 mM, Corning), insulin (0.13%, Millipore, Burlington, Massachusetts, USA), B27 supplement (2%, Gibco), and heat-inactivated FBS (7.5%, Corning, Corning, New York, USA). After cell adhesion, additional plating medium was added. AraC (0.002 mM, Sigma) was added when glia density was 50% to 70% of confluence. Neurons were grown at 37˚C and 5% $CO_2$ in a humidified atmosphere.

To express each of NIR-GECO variants in primary hippocampal neurons and compare their brightness and photostability, the gene encoding NIR-GECO(1,2,2G)-T2A-GFP was constructed using overlap-extension PCR followed by subcloning into pAAV-CAG vector (Addgene no. 108420) using BamHI and EcoRI sites. The gene for miRFP720-P2A-GFP was synthesized de novo by GenScript, based on the reported sequence [13], and cloned into the pAAV-CAG vector. Cultured neurons were transfected with plasmids (1.5 μg of plasmid DNA per well) at 4 to 5 days in vitro (DIV) using a commercial calcium phosphate transfection kit (Thermo Fisher Scientific) as previously described [21,25].

Wide-field fluorescence microscopy of cultured neurons was performed using an epifluorescence inverted microscope (Eclipse Ti-E, Nikon) equipped with an Orca-Flash4.0 V2 sCMOS camera (Hamamatsu) and a SPECTRA X light engine (Lumencor, Beaverton, Oregon, USA). The NIS-Elements Advanced Research (Nikon) was used for automated microscope and camera control. Cells were imaged with a 20× NA 0.75 air objective lens (Nikon) at room

temperature for quantification of brightness or a 40× NA 1.15 for photobleaching experiments (excitation: 631/28 nm; emission: 664LP).

## Field stimulation

Neurons expressing NIR-GECO variants (driven by CAG promoter) were imaged and stimulated in 24-well plates with 300 μL growth medium in each well at room temperature. Field stimuli (83 Hz, 50 V, and 1 ms) were delivered in trains of 1, 2, 3, 5, 10, 20, 40, and 80 via 2 platinum electrodes with a width of 6.5 mm to neurons. Neurons were imaged simultaneously while delivering trains of field stimuli with a 40× NA 1.15, a 631/28-nm LED (Spectra X light engine, Lumencor). Fluorescence was collected through 664LP using a sCMOS camera (Orca-Flash4.0, Hamamatsu) at 2 Hz.

## Ethics statement

All experimental manipulations performed at MIT were in accordance with protocols approved by the Massachusetts Institute of Technology Committee on Animal Care (protocol number: 1218-100-21), following guidelines described in the US National Institutes of Health Guide for the Care and Use of Laboratory Animals. All procedures performed at McGill University were in accordance with the Canadian Council on Animal Care guidelines for the use of animals in research and approved by the Montreal Neurological Institute Animal Care Committee (protocol number: 2015–7728).

## IUE and acute brian slice imaging

In utero electroporation (IUE) was used to deliver the DNA encoding NIR-GECO2 and CoChR or NIR-GECO2G to the mouse brain. Briefly, embryonic day (E) 15.5 timed-pregnant female Swiss Webster (Taconic Biosciences) mice were deeply anesthetized with 2% isoflurane. Uterine horns were exposed and periodically rinsed with warm sterile PBS. A mixture of plasmids pAAV-CAG-NIR-GECO2-WPRE and pCAG-CoChR-mTagBFP2-Kv2.2motif-WPRE or plasmid pAAV-CAG-NIR-GECO2G (at total DNA concentration approximately 1 to 2 μg/μL) were injected into the lateral ventricle of 1 cerebral hemisphere of an embryo. Five voltage pulses (50 V, 50-ms duration, and 1 Hz) were delivered using round plate electrodes (ECM™ 830 electroporator, Harvard Apparatus, Holliston, Massachusetts, USA). Injected embryos were placed back into the dam and allowed to mature to delivery.

Acute brain slices were obtained from Swiss Webster (Taconic Biosciences) mice at postnatal day (P) P11 to P22, using standard techniques. Mice were used without regard for sex. Mice were anesthetized by isoflurane inhalation, decapitated, and cerebral hemispheres were quickly removed and placed in cold choline-based cutting solution consisting of (in mM) 110 choline chloride, 25 $NaHCO_3$, 2.5 KCl, 7 $MgCl_2$, 0.5 $CaCl_2$, 1.25 $NaH_2PO_4$, 25 glucose, 11.6 ascorbic acid, and 3.1 pyruvic acid (339 to 341 mOsm/kg; pH 7.75 adjusted with NaOH) for 2 minutes, blocked, and transferred into a slicing chamber containing ice-cold choline-based cutting solution. Coronal slices (300-μm thick) were cut with a Compresstome VF-300 slicing machine, transferred to a holding chamber with artificial cerebrospinal fluid (ACSF) containing (in mM) 125 NaCl, 2.5 KCl, 25 $NaHCO_3$, 2 $CaCl_2$, 1 $MgCl_2$, 1.25 $NaH_2PO_4$, and 11 glucose (300 to 310 mOsm/kg; pH 7.35 adjusted with NaOH), and recovered for 10 minutes at 34 ˚C, followed by another 30 minutes at room temperature. Slices were subsequently maintained at room temperature until use. Both cutting solution and ACSF were constantly bubbled with 95% $O_2$ and 5% $CO_2$.

Individual slices were transferred to a recording chamber mounted on an inverted microscope (Eclipse Ti-E, Nikon) and continuously superfused (2 to 3 mL/min) with ACSF at room temperature. Cells were visualized through a 10× (0.45 NA) or 20× (0.75 NA) air objective with

epifluorescence to identify positive cells. The fluorescence of NIR-GECO2 or NIR-GECO2G was excited by a SPECTRA X light engine (Lumencor) with 631/28-nm excitation and was collected through a 664LP emission filter, and imaged onto an Orca-Flash4.0 V2 sCMOS camera (Hamamatsu). Optical stimulation of slices expressing NIR-GECO2 and CoChR was performed using a 470-nm LED (M470L3, ThorLabs, Newton, New Jersey, USA) at 0.157 mW/mm$^2$. A 4-AP stimulation was done by adding 4-AP solution to the imaging chamber at a final concentration of 1 mM.

## Imaging of NIR-GECO2 in *C. elegans*

Worms were cultured and maintained following standard protocols [26]. The genes of NIR-GECO2, NIR-GECO2G, HO1, CoChR, and jGCaMP7s for expression in *C. elegans* were codon-optimized using SnapGene codon-optimization tool and synthesized by GenScript. Transgenic worms expressing NIR-GECO2(G) and jGCaMP7s pan-neuronally or NIR-GECO2 in AVA and CoChR-GFP in ASH were generated by injecting the plasmids tag-168::NLS-NIR-GECO2(G)-T2A-HO1(or NIR-GECO2) and tag-168::NLS-jGCaMP7s or plasmids flp-18::NIR-GECO2-T2A-HO1, sra-6::CoChR-SL2-GFP, and elt-2::NLS-GFP into N2 background worms, respectively, picking those with the strongest expression of green fluorescence (in neurons for the pan-neuronal strain and in the gut for optogenetic strain). NLS sequence used in this experiment was PKKKRKV.

Hermaphrodite transgenic worms were picked at L4 stage of development and put onto NGM plates with freshly seeded OP50 lawns 12 to 24 hours before experiments, with or without 100-µM all-trans-retinal (ATR) for optogenetic experiments. Worms were mounted on 2% agarose pads on microscope slides, immobilized with 5 mM tetramisole, covered by a coverslip, and imaged using a Nikon Eclipse Ti inverted microscope equipped with a confocal spinning disk (CSU-W1), a 40×, 1.15 NA water-immersion objective, and a 5.5 Zyla camera (Andor, Belfast, Northern Ireland), controlled by NIS-Elements AR software. To acquire data shown in Fig 3 and S8 Fig, the fluorescence of NIR-GECO2 was imaged with 640-nm excitation provided by a 41.9-mW laser and a 685/40-nm emission filter; jGCaMP7s/GFP fluorescence was imaged with a 488-nm excitation provided by 59.9-mW laser and a 525/50-nm emission filter. Optogenetic stimulation was performed with 488-nm illumination at 20 mW/mm$^2$. For 200 mM NaCl stimulation, worms were imaged using the same optical setup as above, using a microfluidic device that was described previously [20].

For brightness and SBR comparison of NLS-jGCaMP7s and NIR-GECO2, as shown in Fig 4A, 4B and 4D, the plasmids tag168::NLS-jGCaMP7s and tag168::NIR-GECO2-T2A-HO1 were mixed at a ratio of 1:1 before injection. NLS-jGCaMP7s was imaged with 488-nm excitation at a power of 17.2 mW/mm$^2$, and a 525/50-nm emission filter. NIR-GECO2 was imaged with 640-nm excitation at a power of 12.6 mW/mm$^2$,and a 660LP emission filter. All other instrument settings were the same for NLS-jGCaMP7s and NIR-GECO2. The data for brightness from each ROI were averaged by ROI area. SBR was obtained via dividing fluorescence intensity from neurons by averaged autofluorescence from the intestine area. The imaging conditions in Fig 4C and 4E are the same as those in Fig 3. The data for SNR of NLS-jGCaMP7s and NLS-NIR-GECO2 were quantified from spontaneously spiking neurons. SNR was calculated by dividing fluorescence change associated with a spike by the standard deviation (SD) of the baseline fluorescence over the 2-second period immediately before the spike. The imaging conditions in S9 Fig are the same as those in Fig 4.

## Imaging of NIR-GECO2G in *Xenopus laevis* tadpoles

NIR-GECO2G and GCaMP6s were cloned into the pCS2+ vector and the plasmid was linearized with *NotI*. Capped mRNA of NIR-GECO2G and GCaMP6s was transcribed with the SP6

mMessageMachine Kit (Ambion, Thermo Fisher Scientific). The RNA (500 pg of each sample) was injected in 1 blastomere at the 2-cell stage resulting in animals expressing NIR-GECO2G and GCaMP6s protein in 1 lateral half of the animal. The animals were kept at 20˚C until stage 47. Immediately before imaging, the tadpole was paralyzed with pancuronium bromide (1.5 mg/mL in 0.1× MSBH) and embedded in 1% low-melt agarose.

Light-sheet imaging was performed on a Zeiss Z1 located in the McGill Advanced BioImaging Facility. The instrument was equipped with a sCMOS camera (1920 × 1920 pixel PCO. Edge, Kelheim, Germany), excitation lasers with wavelengths of 488 nm (75-mW max. output) and 640 nm (50-mW max. output). For acquisition, we excited the sample with 10% intensity of each wavelength through 5× LSFM excitation objectives (NA = 0.1) resulting in a 4.53-μm thick light sheet. We utilized a water immersion objective for detection (Zeiss 10× PLAN APOCHROMAT, NA = 0.5, UV-VIS-IR, ND = 1.336), allowing the sample to be immersed in 0.1× MBSH. The fluorescence was directed via a dichroic mirror LBF 405/488/640 nm and filtered depending on the probe with 505- to 545-nm bandpass or 660-nm long-pass emission filters.

The imaging data (Fig 5) was acquired with 50-ms integration time and 2.39 seconds cycle time through the volume. The instrument has a short dead time to home the axial position leading to a scanning frequency of 3.0 seconds for the entire volume. The raw data were corrected for drift and rapid movement with the ImageJ plugin TurboReg [27]. The image in Fig 5A is a volume projection of a 45-μm thick volume capturing the spontaneous $Ca^{2+}$ responses in the olfactory bulb. The cells were manually selected if they showed a spontaneous $Ca^{2+}$ response as detected with both NIR-GECO2G and GCaMP6s (Fig 5B). The fluorescence response was measured as the mean fluorescence intensity per cell and normalized by $I_{norm} = (I_m-I_{min})/(I_{max}-I_{min})$. $I_m$ indicates the measured mean value per area and $I_{max}$ and $I_{min}$ the maximal and minimal value measured for the specific ROI. The normalized response of NIR-GECO2G and GCaMP6s of a cell is plotted over time in the graph in Fig 5C.

## Data and image analysis

All images in the manuscript were processed and analyzed using either ImageJ (NIH) or NIS-Elements Advanced Research software (Nikon). Traces and graphs were generated using GraphPad prism 8, Origin (OriginLab, Wellesley, Massachusetts, USA), and Matlab. Data are presented as mean ± SD or mean ± standard error of the mean (SEM) as indicated.

## Supporting information

**S1 Table. Spectral, photochemical, and biochemical properties of NIR-GECO2G, NIR-GECO2, and NIR-GECO1. NIR, near-infrared.**
(DOCX)

**S1 Fig. Sequence alignment of NIR-GECO2, NIR-GECO2G, and NIR-GECO1.** Single amino acid changes relative to NIR-GECO1 are highlighted with a magenta background. PAS domain, GAF domain, linkers, calmodulin, and RS20 are shown as light green, light blue, black, brown, and yellow, respectively. NIR, near-infrared.
(TIFF)

**S2 Fig. Additional in vitro characterization of NIR-GECO variants.** (**a**) $Ca^{2+}$ titration curves of NIR-GECO1, NIR-GECO2, and NIR-GECO2G (center values are the mean and error bars are SD; *n* = 3). (**b**) Photobleaching curves of NIR-GECO variants and mIRFP720 (*n* = 11, 11, 16, and 9 neurons for NIR-GECO1, NIR-GECO2, NIR-GECO2G, and miRFP720, respectively). Mean value (solid line) and SD (shaded areas) are shown. Cells were continuously

illuminated with 631/28 nm at 80 mW/mm$^2$ during the experiment. Images were taken every 5 seconds. (**c**) Representative fluorescence curves of NIR-GECO2G with different imaging rates with 631/28-nm excitation at 2.8 mW/mm$^2$ (this is the excitation intensity that we used for wield-field imaging; exposure time: 100 ms). (**d**) Quantitative data for the photobleaching of NIR-GECO2G and NIR-GECO2 in cultured cells with the same imaging conditions as in **c**. Data are shown as mean ± SD (NIR-GECO2G: $n$ = 70, 71, 62, and 45 cells for 1 Hz, 2 Hz, 5 Hz, and continuous illumination, respectively; NIR-GECO2: $n$ = 26, 36, 35, and 35 cells for 1 Hz, 2 Hz, 5 Hz, and continuous illumination, respectively). (**e**) Brightness of NIR-GECO2G and NIR-GECO2G with co-expression of GCaMP6f in neurons ($n$ = 34 and 56 for NIR-GECOG and NIR-GECO2G with GCaMP6f, respectively; data are shown as mean ± SD). The underlying data for **a**, **b**, **d**, and **e** can be found in S1 Data. NIR, near-infrared; SD, standard deviation. (TIFF)

**S3 Fig. Ca$^{2+}$ indicator prototype based on miRFP.** The mIFP domain of NIR-GECO1 was replaced with miRFP using the same insertion point and linker sequences (i.e., CaM-RS20 was used to replace residues 170 to 177 of miFP or residues 172 to 179 of miRFP; S4 Fig). (**a**) Fluorescence emission spectra of the prototype miRFP-based Ca$^{2+}$ indicator in the presence (5 mM Ca$^{2+}$) and absence (10 mM EGTA) of Ca$^{2+}$. (**b**) Intensity vs. time traces for transfected HeLa cells. Cells were treated with ionomycin/Ca$^{2+}$ to increase cellular Ca$^{2+}$ concentrations and ionomycin/EGTA to deplete cellular Ca$^{2+}$. (**c**) Fluorescent images of miRFP-based Ca$^{2+}$ indicator prototype at time points t1 to t3 (as denoted in **b**). Scale bar, 20 μm; $\lambda_{ex}$ = 650/60 nm and $\lambda_{em}$ = 720/60 nm. Acquisition rate: 0.2 Hz. The underlying data for **a** and **b** can be found in S1 Data. In an effort to develop an improved NIR GECI, we found that the mIFP (engineered from PAS and GAF domain of *Bradyrhizobium* bacteriophytochrome) domain of NIR-GECO1 could be replaced with miRFP [12]. miRFP is another monomeric BV-FP that was derived from *Rps. palustris bacteriophytochrome* and shares 57% amino acid homology with mIFP (S3 and S4 Figs). In principle, the miRFP version on NIR-GECO1 could have served as a template for making improved NIR fluorescent Ca$^{2+}$ indicators due to its higher brightness in mammalian cells relative to mIFP [12]. However, we decided to start our further directed evolution efforts from NIR-GECO1 for 2 reasons. The first reason is that NIR-GECO1 was already optimized and worked well in brain slices, and so starting from it might save time and be associated with a higher chance of success. The second reason is that overexpression of the miRFP-based Ca$^{2+}$ indicator appeared to be toxic to bacteria, and it was challenging for us to incorporate the construct into the bacteria–HeLa screening system that we used for engineering NIR-GECO1 [8]. NIR, near-infrared. (TIFF)

**S4 Fig. Alignment of amino acid sequences of mIFP and miRFP.** Alignment numbering is based on mIFP. The structurally analogous residues between mIFP and mIRFP are highlighted in green. Residues that were replaced by CaM-RS20 to make NIR-GECO1, and the prototype mIRFP-based Ca$^{2+}$ indicator, respectively, are in bold and red. NIR, near-infrared. (TIFF)

**S5 Fig. Testing for possible blue-light-activated photoconversion or photoactivation of NIR-GECO2 and NIR-GECO2G in HeLa cells.** (**a, b**) Averaged fluorescence traces of NIR-GECO2 (**a**, $n$ = 8 cells) and NIR-GECO2G (**b**, $n$ = 9 cells) in response to 100-ms blue light illumination. (**c, d**) Averaged fluorescence traces of NIR-GECO2 (**c**, $n$ = 9 cells) and NIR-GECO2G (**d**, $n$ = 11 cells) in response to 500-ms blue light illumination. Illumination: 470 nm at 6.2 mW/mm$^2$, which is 3.3-fold higher than the intensity used for the experiments

in Fig 2. The underlying data for **a** to **d** can be found in S1 Data. NIR, near-infrared.
(TIFF)

**S6 Fig. Imaging of NIR-GECO2 and NIR-GECO2G in acute brain slices.** (**a**) Wide-field image of a mouse brain slice with co-expression of NIR-GECO2 and CoChR, fluorescence of NIR-GECO2 is shown ($\lambda_{ex}$ = 631/28 nm and $\lambda_{em}$ = 664LP). Scale bar, 50 μm. (**b**) Fluorescence of NIR-GECO2 (acquisition rate 100 Hz) in response to 200-ms blue light stimulation (470/20 nm, 0.157 mW/mm$^2$, indicated by blue bar). The numbers of the traces correspond to the neurons labeled in **a**. (**c**) Single-trial wide-field imaging of 4-aminopyridine (1 mM final concentration) evoked neuronal activity from the 2 representative neurons in brain slices expressing NIR-GECO2 and NIR-GECO2G, respectively ($\lambda_{ex}$ = 631/28 nm and $\lambda_{em}$ = 664LP; acquisition rate: 10 Hz). NIR, near-infrared.
(TIFF)

**S7 Fig. Ca$^{2+}$ imaging in iPSC-CMs using NIR-GECO2G.** (**a**) Representative fluorescence recording of spontaneous and caffeine-evoked Ca$^{2+}$ oscillations using NIR-GECO2G in iPSC-CMs. (**b**) Representative single-trial fluorescence recording of blue light-stimulated Ca$^{2+}$ oscillations (470 nm at a power of 1.9 mW/mm$^2$) using NIR-GECO2G in ChR2 expressed iPSC-CMs. iPSC-CM, induced pluripotent stem cell-derived cardiomyocyte; NIR, near-infrared.
(TIFF)

**S8 Fig. Imaging of *C. elegans* using NIR-GECO2, NIR-GECO2G, and jGCaMP7s.** (**a**) Representative confocal images of worms co-expressing NIR-GECO2-T2A-HO1 and NLS-jGCaMP7s (representative of more than 3 worms). Top, fluorescent image of neurons expressing NLS-jGCaMP7s ($\lambda_{ex}$ = 488-nm laser light, $\lambda_{em}$ = 527/50 nm). Middle, fluorescent image of neurons expressing NIR-GECO2-T2A-HO1 ($\lambda_{ex}$ = 640-nm laser light, $\lambda_{em}$ = 685/40 nm). Bottom, overlay image of green channel and NIR channel. Scale bar, 25 μm. (**b**) Spontaneous Ca$^{2+}$ fluctuation of a representative worm neuron (indicated in **a** by a yellow arrow) co-expressing NIR-GECO2 and NLS-jGCaMP7s. Imaging conditions were identical to the experiments in **a**, acquisition rate: 2 Hz. (**c**) Representative spontaneous Ca$^{2+}$ oscillations of worm neurons reported by NIR-GECO2 (acquisition rate 2 Hz). Imaging conditions were identical to the experiments in **a**. (**d**) Representative confocal images of worms co-expressing NLS-NIR-GECO2G-T2A-HO1 (left) and NLS-jGCaMP7s (right). Imaging conditions were identical to the experiments in **a**. Representative of more than 3 worms. Scale bar, 25 μm. NIR, near-infrared.
(TIFF)

**S9 Fig. Representative images of worms co-expressing NLS-jGCaMP7s (a, $\lambda_{ex}$ = 488-nm laser light, $\lambda_{em}$ = 527/50 nm) and NIR-GECO2 without co-expression of HO1 (b, $\lambda_{ex}$ = 640-nm laser light, $\lambda_{em}$ = 664LP).** Scale bar, 25 μm; representative of *n* = 9 worms. NIR, near-infrared.
(TIFF)

**S1 Video. (Associated to Fig 5).** Imaging of spontaneous neuronal activity with NIR-GECO2G in the olfactory bulb of *Xenopus laevis*.
(MP4)

**S1 Data. Numerical data for Figs 1B–1E, 2, 3B, 3C and 3E and 4 and S2A, S2B, S2D, S2E, S3A and S3B and S5 Figs.**
(XLSX)

## Acknowledgments

We thank the University of Alberta Molecular Biology Services Unit for technical support. We thank Ahmed S. Abdelfattah from Janelia Research Campus for testing NIR-GECO variants and Panagiotis Symvoulidis from MIT for help with data analysis. We also thank Xian Xiao and Hongyun Tang from Westlake University for help with *C. elegans* imaging.

## Author Contributions

**Conceptualization:** Yong Qian, Danielle M. Orozco Cosio, Kiryl D. Piatkevich, Robert E. Campbell.

**Data curation:** Yong Qian.

**Funding acquisition:** Paul W. Wiseman, Edward S. Ruthazer, Edward S. Boyden, Robert E. Campbell.

**Investigation:** Yong Qian, Danielle M. Orozco Cosio, Kiryl D. Piatkevich, Sarah Aufmkolk, Wan-Chi Su, Orhan T. Celiker, Anne Schohl, Mitchell H. Murdock, Abhi Aggarwal.

**Methodology:** Yong Qian, Danielle M. Orozco Cosio, Sarah Aufmkolk, Wan-Chi Su, Yu-Fen Chang, Edward S. Ruthazer, Robert E. Campbell.

**Project administration:** Paul W. Wiseman, Edward S. Ruthazer, Edward S. Boyden, Robert E. Campbell.

**Supervision:** Yu-Fen Chang, Paul W. Wiseman, Edward S. Ruthazer, Edward S. Boyden, Robert E. Campbell.

**Validation:** Yong Qian.

**Writing – original draft:** Yong Qian, Robert E. Campbell.

**Writing – review & editing:** Yong Qian, Danielle M. Orozco Cosio, Kiryl D. Piatkevich, Sarah Aufmkolk, Wan-Chi Su, Orhan T. Celiker, Anne Schohl, Mitchell H. Murdock, Abhi Aggarwal, Yu-Fen Chang, Paul W. Wiseman, Edward S. Ruthazer, Edward S. Boyden, Robert E. Campbell.

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
