## [Editor Report · Decision Letter 0]

17 Apr 2020

Dear Dr Campbell, 

Thank you for submitting your manuscript entitled "Improved genetically encoded near-infrared fluorescent calcium ion indicators for in vivo imaging" for consideration as a Methods and Resources article by PLOS Biology.

Your manuscript has now been evaluated by the PLOS Biology editorial staff, as well as by an Academic Editor with relevant expertise, and I am writing to let you know that we would like to send your submission out for external peer review.

Please re-submit your manuscript within two working days, i.e. by Apr 21 2020 11:59PM.

Kind regards,

Gabriel Gasque, Ph.D.,

Senior Editor

PLOS Biology

---

## [Decision Letter · Decision Letter 1]

4 Jun 2020

Dear Dr Campbell,

Thank you very much for submitting your manuscript "Improved genetically encoded near-infrared fluorescent calcium ion indicators for in vivo imaging" for consideration as a Methods and Resources article at PLOS Biology. Your manuscript has been evaluated by the PLOS Biology editors, by an Academic Editor with relevant expertise, and by three independent reviewers.

In light of the reviews (below), we will not be able to accept the current version of the manuscript, but we would welcome re-submission of a much-revised version that takes into account the reviewers' comments. We cannot make any decision about publication until we have seen the revised manuscript and your response to the reviewers' comments. Your revised manuscript is also likely to be sent for further evaluation by the reviewers.

We expect to receive your revised manuscript within 2 months. 

**IMPORTANT - SUBMITTING YOUR REVISION**

Your revisions should address the specific points made by each reviewer. As you will see, the reviewers are overall very supportive. However, reviewers 2 and 3 indicate that additional experiments and controls would provide very useful information to the community. Considering this is a Resource, and having discussed these requests with the Academic Editor, we think that to strengthen the utility of these probes, you should address these points with additional data.

Please submit the following files along with your revised manuscript:

*Re-submission Checklist*

*Published Peer Review*

*PLOS Data Policy*

*Blot and Gel Data Policy*

Sincerely,

Gabriel Gasque, Ph.D., 

Senior Editor

PLOS Biology

REVIEWS:

Reviewer #1: In general, compared to the conventional fluorescent calcium ion optogenetic tools and indicators functioned under visible wavelength light irradiation, the NIR genetically-encoded counterparts provide great advantages of reduced light toxicity, less background cross-talk with visible-light as well as decreased scattering and absorption in mammalian tissues. However, several disadvantages including lower brightness and limited fluorescence response remained the concerns to hamper their further applications for imaging of neuronal activity in vitro and in vitro. To overcome these technical barriers, in this study, qiao and co-authors genetically encoded two ca2+ indicators, NIR-GECO2 and NIR-GECO2G, to provide substantial improvements of neuronal imaging of Ca2+ dynamics in cultured cells, mouse brain slices, and C. elegans and Xenopus laevis in vivo. Even though some new challenges like less brightness, slower kinetics, and faster photobleaching still need to be further improved. Apparently, the authors have carried out extensive experiments and the manuscript has been well prepared. The referee thus supports the publication of this interesting study in Plos Biology as the current stage. 

Reviewer #2: In this manuscript by the Campbell lab (Qian et al), the authors described new variants of genetically encoded near-IR Ca2+ indicator NIR-GECO2 and NIR-GECO-2G. Compared to previously developed NIR-GECO1, there was not significant improvement in brightness or photo-stability, but they have 2-4 fold higher affinity to Ca2+. This increased dF / Fo of Ca2+ signals in neurons by 2-3 times. They showed that the new reagents are useful for Ca2+ imaging in acute slices, as well as in vivo imaging in C-elegans or Xenopus Laevis.

Overall the improvement in Ca2+ sensitivity seems to be significant, and thus the reagent should be useful for the community. One significant concern is the lack of control experiments for the photo-conversion by blue light illumination (Fig. 2). Since they used channel rhodopsin to test the sensitivity of NIR-GECO in these experiments, I think it is important to report to what degree NIR-GECO shows photo-conversion by blue light. For the community, the degree of compatibility with channel rhodopsin would be a great interest too. 

Reviewer #3: Qian et al. reported development and in vivo demonstration of two improved near-infrared fluorescence calcium indicators. Compared to the parent sensor, NIR-GECO1, these two variants displayed increased affinity to calcium and about 3 to 4-fold higher ∆F/F in response to single-field stimuli in dissociated neuronal culture or light-stimuli (100ms) in the brain slice. Other intrinsic properties remain similar to NIR-GECO1. The improved sensitivity permitted populational imaging of optogenetically triggered or spontaneous neuronal activity in c.elegans and in Xenopus laevis, thus representing a substantial improvement compared to NIR-GECO1. 

The scope and novelty are appropriate to PLOS Biology. However, the authors should address/discuss a few points about in vivo utility of these sensors. 

1. One of potential advantages using NIR-based probes is the depth penetration. In Fig 3 and 4, the authors nicely demonstrated calcium signals detected by NIR-GECO. It will significantly strengthen the utility of these probes if the authors compare the signal-to-noise ratio of NIR-GECO and jGCaMP/GCaMP and 3D volume/optical dissection depth possibly increased by using longer-wavelength. 

2. The photostability is a major barrier for red-shifted calcium indicators. What is the highest acquisition rate of recording permitted without significant photobleaching, 2Hz? Can the same neuronal population be imaged repeatedly without significant photobleaching? 

3. In Fig 3, the authors co-expressed HO1 to enhance the BV level in neurons. However, the authors did not mention the fold-increase in terms of brightness without HO1 co-expression. 

4. It is not clear whether co-expression with jRCaMP or GCaMP6s would influence the expression level of NIR-GECO2/2G. In Fig 3a and 4b, the express patterns of two sensors are not exactly the same. Can authors clarify whether co-expression of another sensor would influence the expression and performance of NIR-GECO? 

5. In Fig 3b and Fig 4c, the waveform of jGCaMP7 or GCaMP6s is not exactly the same with that of NIR-GECO2. For example, there is a mall transient right before the third puff of NaCl detected by jGCaMP7, but not by NIR-GECO2. Is this due to the sensitivity of the sensor. In Fig 4c, the fluorescence signal of NIR-GECO2G seems to lag behind GCaMP6s. it requires some discussion about the discrepancy of calcium signals detected by different sensors. 

6. As there is no in vivo demonstration in rodent, I assume it remains challenging to image neuronal activity with NIR-GECO2 in rodents. Can the authors discuss the challenges/limitations preventing the use of these variants in rodents? 

7. The authors nicely demonstrated that the sensor can be targeted to nucleus in Fig 3. Though it may not be useful for in vivo imaging, it will be useful to use 2 or 3 colors to detect calcium transients in various subcellular locations. This could be a good discussion point about usefulness of these probes.

---

## [Decision Letter · Decision Letter 2]

3 Oct 2020

Dear Dr Campbell,

Thank you for submitting your revised Methods and Resources article entitled "Improved genetically encoded near-infrared fluorescent calcium ion indicators for in vivo imaging" for publication in PLOS Biology. I have now obtained advice from the original reviewers 2 and 3 and have discussed their comments with the Academic Editor. You will note that reviewer 3, Lin Tian, has revealed her identity. Please accept my apologies for the delay in communicating this decision to you.

We're delighted to let you know that we're now editorially satisfied with your manuscript. However before we can formally accept your paper and consider it "in press", we also need to ensure that your article conforms to our guidelines. A member of our team will be in touch shortly with a set of requests. As we can't proceed until these requirements are met, your swift response will help prevent delays to publication. Please also make sure to address the data and other policy-related requests noted at the end of this email.

- a cover letter that should detail your responses to any editorial requests, if applicable

*Copyediting*

*Published Peer Review History*

*Early Version*

Sincerely,

Gabriel Gasque, Ph.D.,

Senior Editor,

ggasque@plos.org,

PLOS Biology

ETHICS STATEMENT:

-- Please include in your manuscript the ID numbers of the protocols approved by the Massachusetts Institute of Technology Committee on Animal Care and by the the Montreal Neurological Institute Animal Care Committee.

DATA POLICY:

Note that we do not require all raw data. Rather, we ask for all individual quantitative observations that underlie the data summarized in the figures and results of your paper. For an example see here: http://www.plosbiology.org/article/info%3Adoi%2F10.1371%2Fjournal.pbio.1001908#s5

These data can be made available in one of the following forms:

Regardless of the method selected, please ensure that you provide the individual numerical values that underlie the summary data displayed in the following figure panels: Figures 1b-e, 2a-d, 3bce, 4a-e, S2abde, S3ab, and S5a-d.

Please also ensure that each figure legend in your manuscript includes information on where the underlying data can be found and that your supplemental data file/s has/have a legend.

Reviewer remarks:

Reviewer #2: All concerns are addressed. 

Reviewer #3, Lin Tian: The authors have thoroughly addressed my conners. I believe NIR-GECO2 is a valuable tool to the field and should be published and disseminated asap.

---

## [Editor Report · Decision Letter 3]

29 Oct 2020

Dear Dr Campbell,

On behalf of my colleagues and the Academic Editor, Polina V. Lishko, I am pleased to inform you that we will be delighted to publish your Methods and Resources in PLOS Biology. 

PRODUCTION PROCESS

Before publication you will see the copyedited word document (within 5 business days) and a PDF proof shortly after that. The copyeditor will be in touch shortly before sending you the copyedited Word document. We will make some revisions at copyediting stage to conform to our general style, and for clarification. When you receive this version you should check and revise it very carefully, including figures, tables, references, and supporting information, because corrections at the next stage (proofs) will be strictly limited to (1) errors in author names or affiliations, (2) errors of scientific fact that would cause misunderstandings to readers, and (3) printer's (introduced) errors. Please return the copyedited file within 2 business days in order to ensure timely delivery of the PDF proof. 

If you are likely to be away when either this document or the proof is sent, please ensure we have contact information of a second person, as we will need you to respond quickly at each point. Given the disruptions resulting from the ongoing COVID-19 pandemic, there may be delays in the production process. We apologise in advance for any inconvenience caused and will do our best to minimize impact as far as possible.

EARLY VERSION

PRESS 

Kind regards,

Vita Usova

Publication Editor, 

PLOS Biology

on behalf of

Gabriel Gasque,

Senior Editor

PLOS Biology